# Identifying the Active Sites of Heteroatom Graphene as a Conductive Membrane for the Electrochemical Filtration of Organic Contaminants

**DOI:** 10.3390/ijms232314967

**Published:** 2022-11-29

**Authors:** Meilan Pan, Junjian Li, Bingjun Pan

**Affiliations:** College of Environment, Zhejiang University of Technology, Hangzhou 310014, China

**Keywords:** electrochemical filtration, conductive membrane, heteroatom graphene, S-doped, N-doped

## Abstract

The dopants of sulfur, nitrogen, or both, serving as the active sites, into the graphitic framework of graphene is an efficient strategy to improve the electrochemical performance of electrochemical membrane filtration. However, the covalent bonds between the doped atoms and the substrate that form different functional groups have a significant role in the specific activity for pollutant degradation. Herein, we found that the singly doped heteroatom graphene (NG and SG) achieved superior removal efficiency of pollutants as compared with that of the double doped heteroatom graphene (SNG). Mechanism studies showed that the doped N of NG presented as graphitic N and substantially increased electron transfer, whereas the doped S of SG posed as -C-SOx-C- provided more adsorption sites to improve electrochemical performance. However, in the case of SNG, the co-doped S and N cannot form the efficient graphitic N and -C-SOx-C- for electrochemical degradation, resulting in a low degradation efficiency. Through the fundamental insights into the bonding of the doped heteroatom on graphene, this work furnishes further directives for the design of desirable heteroatom graphene for membrane filtration.

## 1. Introduction

Graphene with fascinating properties has been changing the landscape of many fields in science and technology [1,2,3,4], particularly in terms of condensed matter physics, electronics, energy storage and conversion, and environmental remediation [5,6,7]. Pristine graphene, a single-atom-thick layer of sp2 bonded carbon atoms tightly packed into a 2D honeycomb lattice, is largely understood and well-recognized through extensive research in the past years [8]. However, because of the lack of intrinsic bandgap, pristine graphene is unfavorable for catalysis and is greatly limited concerning practical applications [9]. Many strategies are proposed to modify its intrinsic properties and raise new application opportunities. Notably, doping graphene with various heteroatoms (e.g., nitrogen, boron, sulfur, phosphorus, nitrogen/sulfur, or nitrogen/boron) by substituting or covalently bonding, can inevitably cause structural and electronic distortions [10,11,12], leading to the alteration of pristine graphene properties, such as thermal stability, charge transport [13], Fermi level, bandgap, localized electronic state, spin density, and optical characteristics [14].

Heteroatom graphene, by doping foreign atoms with varying configurations and doping levels, endows graphene with a wide spectrum of new properties to achieve superior electrochemical performances [15,16,17]. For instance, N-doping can induce a bandgap near the Dirac point by suppressing the nearby density of states (DOS) [18], thereby endowing graphene with n-type semiconducting properties, which is advantageous in reactions such as ORR and H_2_O_2_ reduction [19,20]. Unlike the N-doping effect, although a negligible polarization (or charge transfer) exists in the C-S bond, because of the similar electronegativities of S (2.58) and C (2.55) [21], the substitution of S atoms for C atoms at the graphene edges results in higher charge and spin densities, which can subsequently enhance graphene conductivity due to the stronger electron donor ability and more effective reduction by S doping [22,23,24]. Therefore, varying the covalent bonds of the doped atoms to form different functional groups has a significant role in the specific activity.

Electrochemical membrane filtration, coupling membrane filtration with electrocatalysis, can not only enhance the electrochemical degradation efficiency but also effectively eliminate membrane fouling. In the electrochemical filtration process, a conductive membrane serves as the membrane filter and the electrochemical anode [25,26]. Especially, a graphene membrane possessing abundant two-dimensional nanochannels in-between stacked nanosheets is particularly attractive, because it offers high tunability in terms of surface chemical and electrochemical properties owing to the dual functions of the membrane and electrode [27,28]. Heteroatom graphene can be considered an environmentally friendly conductive membrane material for oxide and reductive in situ degradation pollutants in the electrochemical filtration process [29,30,31]. However, the intrinsic properties of the heteroatom graphene conductive membrane in electrochemical filtration are still not well recognized [32,33].

In this work, three kinds of heterogeneous graphene, including nitrogen-doped graphene (NG), sulfate-doped graphene (SG), and sulfate/nitrogen-co-doped graphene (SNG), were successfully synthesized via a hydrothermal method. The results showed that the singly doped heteroatom graphene (NG and SG) exhibited superior removal efficiency for pollutants as compared with that of the double doped heteroatom graphene (SNG). The experimental and theoretical results demonstrate that the doped N of NG presented as graphitic N and substantially increased electron transfer, and the doped S of SG posed as -C-SOx-C- and provided more adsorption sites to improve the electrochemical performance. However, the co-doped S and N cannot form the efficient graphitic N and -C-SOx-C- in SNG for electrochemical degradation, resulting in a low degradation efficiency. More mechanism studies were performed and revealed that various functional groups formed in singly heteroatom graphene, which played a critical role in the electrochemical filtration of contaminants. 

## 2. Results and Discussion

### 2.1. Preparation and Characterization of Heteroatom Graphene

In a typical synthetic process [34], GO solution (ca. 1 mg mL^−1^) serves as the starting material to prepare O doped (OG, Figure 1a), S/O doped (SG), N/O doped (NG), and S/N/O doped graphene (SNG) under different hydrothermal condition. Scanning electron microscopy (SEM) and transmission electron microscopy (TEM) images can reveal their typical graphene characteristics (Figure 1b,g). X-ray diffraction (XRD, Figure 2a) shows a wide reflection (2θ = 27.0°) corresponding to graphite, and S/N/S; N doped graphene had a significant shift as compared with that of OG due to the large size of the doped (S or N) atoms. The FTIR spectra (Figure 2b) shows the obvious stretching vibration of C-N (1250 cm^−1^) on NG and SNG, C-S bond (800 cm^−1^) and S=O bond (1400 cm^−1^) on SG and SNG, and C-O bond (1020 cm^−1^) on all samples, demonstrating the success dopant of N/S into the structure of graphene [35,36,37]. UV-vis spectra (Figure 2c) reveal the characteristic peaks of the π-π* transition of OG around 275 nm. Obviously, the absorbance bands of NG/SG/SNG were broader and shifted rightward to ca. 278 nm, suggesting the incorporation of the S/N atom into graphene [34,38].

The chemical composition and the elemental state of sulfur and nitrogen dopants in different materials were investigated by elemental analysis (EA) (Table 1) and XPS (Table 2). The element analysis showed that the ca. 2.4 at% of S was doped into SG, and the ca. 4.3 at% of N was doped into NG, but the S dopant in SNG increased to 6.6 at% and that of N decreased to 6.3 at%. N 1s and S 2p specific binding energies confirm the formation of covalent bonds for the doped N and S in Figure 3. The high-resolution N 1s XPS spectra can be deconvoluted into three types of nitrogen species; namely, pyridinic (~398.6 eV), pyrrolic (~400.5 eV), and graphitic (~401.3 eV) nitrogen [39]. Both NG and SNG were mainly presented as pyrrolic nitrogen, but NG had more graphic nitrogen (29.3%) than that of SNG (15.7%) (Table 2). With respect to S 2p spectra, S 2p_3/2_ and S 2p_1/2_ doublet located at 163.1 and 164.5 eV spin-orbit levels with an energy separation of 1.0 eV and an intensity ratio of about 1:2 [40,41]. These signals can be attributed to the formation of C=S and C-S bonds, further confirming the dopant of the graphene network by sulfur. The peaks at 168.4–170.1 eV corresponding to -C-SO_x_-C- possibly originate from the oxidic sulfur species. Table 1 and Appendix A showed that the S in SG mostly presented as -C-SO_x_-C- bonds, but the S in SNG was mostly presented as -C-S-C- bonds. These results identified the different covalent bonds with the comparison of singly doped and double doped heteroatom graphene.

### 2.2. Electrochemical Performance in Membrane Filtration 

The hybrid graphene membrane was fabricated by assembling the heteroatom graphene onto a membrane; this served as the anodic material to *on-line* the degradation pollutants electrochemically (MB and RhB). Figure 4a and Appendix A show the adsorption and degradation curves of MB, in which MB was adsorbed on different graphene membranes and then was degraded at an applied potential (3 V). The MB adsorption capacities of the four graphene materials decreased in the following order: NG (261.2 mg/g) > OG (182.7 mg/g) > SNG (123.3 mg/g) > SG (106.6 mg/g) (Appendix A). SG, NG, and SNG, as the conductive membranes, exhibited superior *on-line* degradation efficiency of MB; especially, rapid degradation kinetics by SG and NG were observed, and their degradation efficiencies of ca. 94.7% and 92.8% were reached in 15 min, whereas only 73.1% was achieved by OG. On the other hand, although SNG also showed decent degradation efficiency (90.6%), its relatively poor degradation kinetics implies worse electrochemical performance as compared with those of SG and NG. Furthermore, results at lower potentials (1 and 2 V) showed more differences on these doped graphene membranes, where the degradation efficiencies follow the order: SG > NG > SNG. Thus, it can be concluded that NG and SG are more efficient than SNG as conductive membranes for the *on-line* degradation of MB in a filtration system. 

The *on-line* degradation of RhB is performed in this membrane filtration system. The adsorption capacities follow the order: NG (169.9 mg/g) > SNG (141.4 mg/g) > OG (128.6 mg/g) > SG (93.4 mg/g) (Appendix A). As compared with the results of the MB degradation, NG is more favorable for RhB degradation with a degradation efficiency of 92.1%. The degradation efficiency of SG (81.9%) is higher than that of OG (65.6%) but is still lower than that of NG, indicating that the doped N has a higher activity than the doped S. These results suggest that singly doped heteroatom (S or N) graphene is more favorable for electrochemical membrane filtration than double doped heteroatom graphene (SNG).

The specific activities of OG, NG, SG, and SNG were further analyzed to compare their electrocatalytic activities. The specific activity was calculated by normalizing the removal efficiency with the electrochemical active surface area (*ECSA*):(1)$$Specific\;activity=\frac{\left(C_{0}-C\right)\times J}{ECSA\times m}$$
where *C* is the effluent concentration of pollutants, *C*_0_ is the influent concentration of pollutants, *ECSA* is the electrochemical active surface area, *J* is the flow velocity, and *m* is the mass of the membrane.

Figure 4c,f show that SG and NG gave much higher specific activities than that of SNG, that is, 0.32 mg (MB) m^−2^ (SG) > 0.30 mg (MB) m^−2^ (NG) > 0.17 mg (MB) m^−2^ (SNG); 0.12 mg (RhB) m^−2^ (SG) > 0.21 mg (RhB) m^−2^ (NG) > 0.08 mg (RhB) m^−2^ (SNG). These results further confirmed that singly doped heteroatom graphene (SG and NG) is favorable for electrochemical membrane filtration as compared with double doped heteroatom graphene (SNG).

### 2.3. Identification of Active Sites 

The incorporation of an S or N atom into the graphitic basal plane can create active sites on the adjacent carbon atoms; thus, facilitating adsorption and redox reactions. We further investigate the roles of different covalent bond configurations for reaction. Figure 5a,c show the dependence of electrochemical activity on the percentage of N and S dopants. Remarkably, the content of pyridinic and pyrrolic N was negatively correlated to the degradation efficiency, whereas the graphitic N content was positively correlated to the degradation efficiency, indicating that graphitic N served as the active sites for the electrochemical filtration of pollutants. Generally, pyrrolic and pyridinic N have a non-bonding electron pair that favors the adsorption of oxygen or pollutants [42], whereas graphitic N without non-bonding electrons could enhance conductivity, which favors electron transport during the charge and discharge process [43]. Thus, the electron transfer on N doped graphene is critical for the removal of pollutants, implying that increasing the conductivity of the membrane is an efficient strategy for electrochemical degradation. The transfer rate of reaction electrons passing between the reaction interfaces and/or along the graphene surface can be assessed by EIS. EIS was performed in the electrochemical oxidation of BPA on NG, SG and SNG (Figure 5b). Data were simulated using *Zview* software, and the results showed that the R_ct_ values of NG (7.12 Ω) are much smaller than those of SNG (12.81 Ω) and SG (11.71 Ω). By augmenting with the more graphitic N, NG reveals a higher electron transfer performance in comparison with SNG, verifying that the non-bonding electrons of graphitic N enhance conductivity. 

Similarly, the dependence of electrochemical activity on the content of S dopants showed the dominate role of the covalent -C-SO_x_-C- bond rather than the -C-S-C- bond for electrochemical degradation. Unlike the effect of N doping, a negligible charge transfer exists in the -C-S-C- bond because of the similar electronegativity of S (2.58) and C (2.55) [17]. In general, SG is more resistive than pristine graphene because of the free carrier trapping caused by the S and O functionalities [4]. However, the trapped free carriers on -C-SO_x_-C- breaks the electroneutrality of SG and creates favorable positive charged sites for the adsorption of oxygen or pollutants. In particular, the vibrations of S-O of SG exhibited clear shifts from 1012.4 cm^−1^ to 1039.4 cm^−1^ (Figure 5d), verifying that the efficient adsorption primarily occurred between the -SO_x_ groups of SG and the pollutants. However, the favorable covalent bond cannot be formed on the surface of SNG in abundance.

### 2.4. Electrochemical Activities of Heteroatom Doped Graphene

The above experimental results clearly show that the graphitic N and -C-SO_x_-C- on SG and SNG are attributed to the high degradation efficiency toward electrochemical filtration; moreover, the electron transfer behaviors determine the electrochemical performance. Accordingly, we used classical electrochemical probes to verify the possible different electron transfer behaviors of heteroatom graphene. Ferrocyanide ([Fe (CN)_6_]^3−/4−^) is a classic anodic molecular probe used as an ideal outer-sphere electron transfer and has negligible adsorption. In the CV analyses (Figure 6a,c), the redox currents and peak-to-peak separation on NG (∆Ep = 139 mV), SG (∆Ep = 127 mV), and SNG (∆Ep = 166 mV) at a scan rate of 50 mV s^−1^ using 1 mM of [Fe (CN)_6_]^3−/4−^ (Appendix A), indicate the ideal one-electron Nernst system with diffusion control due to convective mass transfer enhancements. Although this probe is somewhat sensitive to the surface compositions of carbon electrodes, it is generally accepted that the coordination shell of [Fe (CN)_6_]^3−/4-^ remains intact on carbon surfaces in electron transfer. The [Fe (CN)_6_]^3−/4−^ redox peak positions and charge transfer resistances suggests that the activity of the doped N on NG and SNG is dependent on the species of N, in which the graphic N is the dominate species to enhance electron transfer for electrochemical filtration. For SG and SNG, they have similar electron transfer behaviors. 

We further compared the electron transfer behaviors of SG and SNG with the Fe^2+/3+^ (H_2_O) redox system, which is known to be very sensitive to the electrode surfaces via inner-sphere electron transfer. Unlike that of [Fe(CN)_6_]^3−/4−^, the redox kinetics of Fe^2+/3+^(H_2_O) greatly depends on the chemical bonding with the surface active sites of graphene electrodes, which approximates the electron transfer mechanism in the electrocatalytic reactions [44]. The redox currents of Fe^2+/3+^(H_2_O) was 0.64 mA with ∆Ep of 108 mV on SG (Figure 6b), whereas the redox currents decreased to 0.26 mA and 0.22 mA, and ∆Ep drastically increased to 352 mV and 507 mV for NG and SNG, respectively. Combining this with the structural analysis, the distinct electron transfer kinetics of Fe^2+/3+^(H_2_O) implies that the -C-SO_x_-C- covalent bond, through providing favorable adsorption sites, improved the electrochemical filtration by facilitating electron transfer at the graphene interface (Figure 6c). 

## 3. Materials and Methods

### 3.1. Materials and Reagents

All chemicals, including methylene blue (MB), rhodamine B (RhB), sodium sulfate, ethanol, potassium ferricyanide, iron sulfate, potassium chloride, isopropanol (IPA), and dimethylformamide (DMF), are reagent grade and were purchased from Sigma-Aldrich. The graphite flakes (44 μm of average particle diameter, 99.95% of purity) are from Qingdao Hengdeli Graphite Co., Ltd. (Qingdao, China).

### 3.2. Synthesis of Heteroatom Graphene

Graphene oxide (GO) was synthesized from pristine graphite flakes through a modified Hummers’ method. Briefly, graphite powder (6 g) was mixed with K_2_S_2_O_8_ (2 g) and P_2_O_5_ (2 g) in 98% H_2_SO_4_ (20 mL) and was then heated at 80 °C for 10 h. The resultant mixture was dispersed in 98% H_2_SO_4_ (50 mL) to further react with KMnO_4_ (6 g) at 40 °C for 2 h to form GO. Finally, GO was washed with a large amount of DI water to remove the residual ions and then uniformly dispersed in water ultrasonically to obtain a nearly homogenous graphene solution. GO was dispersed in water (1 mg mL^−1^), and the suspension was transferred into a Teflon-lined stainless autoclave for thermal treatment at 150 °C for 3 h to obtain the O doped graphene (OG). To obtain heteroatom graphene [nitrogen-doped (NG), sulfate-doped graphene (SG), and sulfate/nitrogen-co-doped graphene (SNG)], urea, sodium hydrosulfide, and thiourea were added into the GO solution [N/S:C (GO) = 50:1], respectively. The other procedures were the same as those of the OG.

### 3.3. Preparation of Electrochemical Membrane Filtration Device

The filtration membrane of the heteroatom-doped graphene was prepared by dispersing 20 mg of heteroatom graphene in isopropanol (IPA) by probe sonication (Nanjing Betty, Nanjing, China) at an applied power of 400 W L^−1^ for 20 min. Then, the prepared dispersion (30 mL) was filtered through a 5-μm polytetrafluoroethylene membrane. The membrane was subsequently washed with 100 mL of ethanol (EtOH), 100 mL of a 1:1 H_2_O/EtOH mixture, and 100 mL of H_2_O to remove the residual IPA. The prepared membrane was loaded onto a tailored filtration casing to explore the electrochemical degradation performance using methylene blue (MB) or rhodamine B (RhB) as the model pollutant in 10 ± 1.0 mg L^−1^ Na_2_SO_4_ unless otherwise specified. During the operation, the sample membrane was used as the anode, electrically connected via a titanium ring and wire to a DC power supply. The appropriate influent solution was then peristaltically pumped (Longer) through the sample membrane at a flow rate of 0.5 ± 0.1 mL min^−1^. Sample aliquots were collected directly from the filtration casing outlet and analyzed immediately, as described in our previous studies [45]. The concentrations of MB and RhB were measured by an ultraviolet-visible (UV-vis) spectrophotometer (Agilent, Santa Clara, CA, USA).

### 3.4. Material Characterizations

The microscopic features and morphology of the samples were characterized by scanning electron microscope (SEM, Zeiss Sigma 300, Oberkochen, Germany) and transmission electron microscopy (TEM, EI Tecnai F20, Hillsboro, OR, USA). X-ray diffraction (XRD) analysis of the samples was performed with an X-ray diffractometer (Shimazduo 6000, Kyoto, Japan) using Cu Kα radiation at a scanning rate (2θ/min) of −2°. The surface functional groups were observed by X-ray photoelectron spectroscopy (XPS) and Fourier transform infrared spectroscopy (FT-IR). The XPS data were collected with a Thermo Scientific K-Alpha with a resolution below 0.2 eV, and the C 1a peak spectra were analyzed using XPS Peak version 4.1. The FT-IR spectra were recorded on a Thermo Fisher spectrometer (Nicolet iS5, Madison, WI, USA) in the 4000–5000 cm^−1^ region with a resolution of 4 cm^−1^ in transmission mode. 

### 3.5. Electrochemical Characterizations

For the electrochemical characterization, a potentiostat was used to perform cyclic voltammetry (CV) with a CHI 660E electrochemical workstation (CHI, Austin, TX, USA). The sample was employed as the working electrode, and a stainless-steel cathode was used as the counter electrode with an Ag/AgCl electrode as the reference electrode. All anode potentials listed in the text and figures were with respect to the Ag/AgCl reference electrode. CV was performed under different probes including potassium ferricyanide (K_2_[Fe(CN)_6_]) and ferric sulfate (Fe_2_(SO_4_)_3_). The resultant data were modeled with Nyquist plots.

With a similar composition and structure, the electrochemical active surface area (*ECSA*) of an electrode material is proportional to its electrochemical double-layer capacitance ($C_{dl}$). It is measured by CV in the non-Faraday region at different scan rates (*V_b_*) of 10, 20, 50, 80, 100, 200, 300, 500 and 700 in the three-electrode system. Then the $C_{dl}$ was estimated by plotting the Δj = $\left(j_{a}-j_{c}\right)$ at 0.4 V vs Ag/AgCl as a function of the scan rate. It can be calculated according to the equation [46]: (2)$$C_{dl}=\frac{d\left(\Delta j\right)}{2dV_{b}}$$

The *ECSA* can be calculated from $C_{dl}$ as follows:(3)$$ECSA=\frac{C_{dl}}{C_{s}}$$
where *C_s_* is the specific capacitance of a flat surface with 1 cm^2^ of surface area. Appendix A show the *ECSA* values of OG, NG, SG and SNG.

## 4. Conclusions

In summary, we found that singly doped heteroatom graphene (NG and SG) achieved superior removal efficiency for pollutants as compared with that of double doped heteroatom graphene (SNG). Mechanism studies showed that the doped N on NG presented as graphitic N to increase electron transfer, whereas the doped S on SG exhibited as -C-SO_x_-C- to serve as adsorption sites and, thus, improved the electrochemical performance. Unlike the case of NG and SG, the doped S and N on SNG cannot form the efficient graphitic N and -C-SO_x_-C- for electrochemical degradation, resulting in a low degradation efficiency. Through the fundamental insights gained into the bonding of the doped heteroatom on graphene, this work furnishes further directives for the design of heteroatom graphene for membrane filtration.

## Figures and Tables

**Figure 1 ijms-23-14967-f001:**
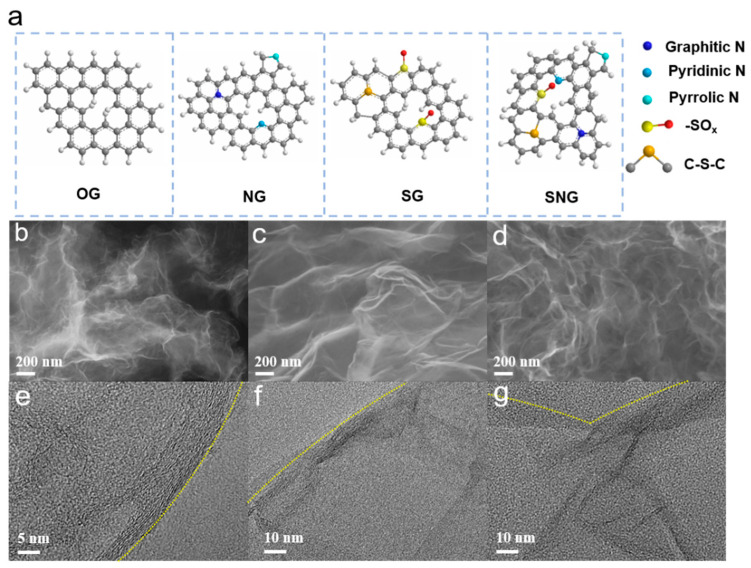
(**a**) Schematic illustrating the structures of OG, NG, SG and SNG. SEM (**b**–**d**) and TEM (**e**–**g**) images of NG, SG, and SNG.

**Figure 2 ijms-23-14967-f002:**
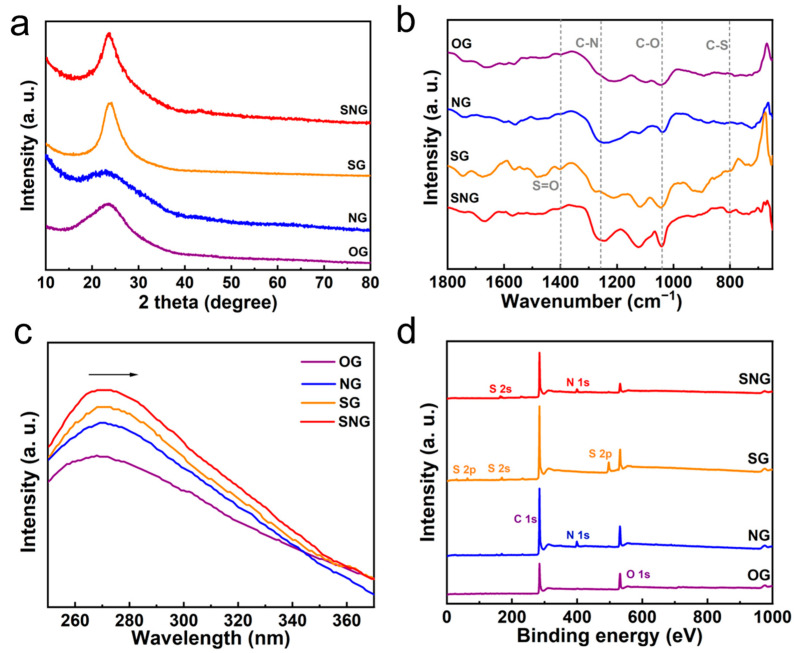
XRD (**a**), FTIR (**b**), UV-vis (**c**), and XPS (**d**) patterns of heteroatom doped graphene (OG, NG, SG, and SNG).

**Figure 3 ijms-23-14967-f003:**
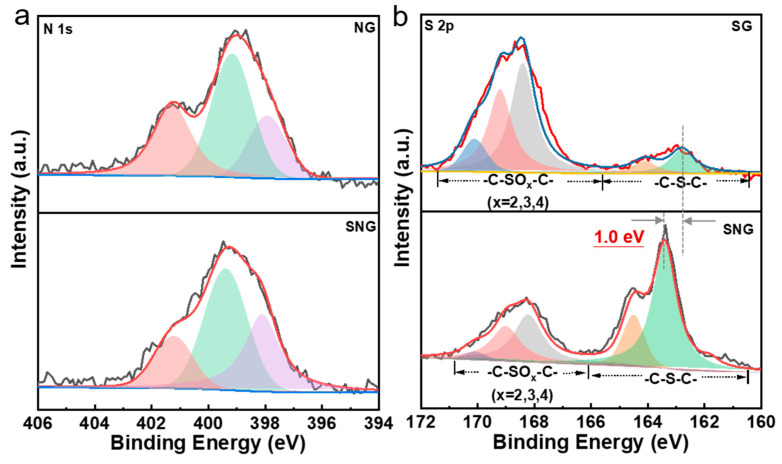
XPS spectra of NG, SG, and SNG. (**a**): N 1s of NG and SNG. (**b**): S 2p of SG and SNG.

**Figure 4 ijms-23-14967-f004:**
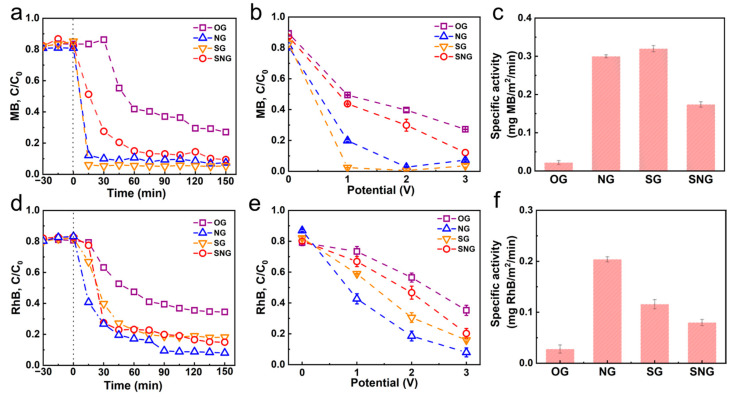
(**a**) Electrochemical degradation of MB at 3V. (**b**) Electrochemical oxidation of MB under different voltage. (**c**) The specific activity of the sample for MB removal. (**d**) Electrochemical desorption and oxidation of RhB at 3V. (**e**) Electrochemical oxidation of RhB under different voltages. (**f**) The specific activity of the sample for RhB removal.

**Figure 5 ijms-23-14967-f005:**
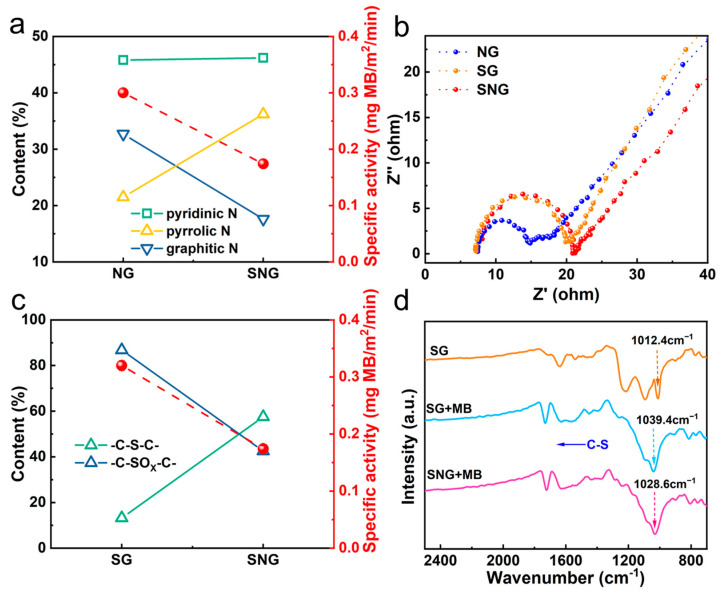
The dependence of specific electrochemical activity on the species of N (**a**) and S (**c**). (**b**) EIS and simulated plots of NG, SG and SNG in Fe (CN)_6_^3−/4−^ solution. (**d**) FTIR spectra of SG, SG + MB and SNG + MB.

**Figure 6 ijms-23-14967-f006:**
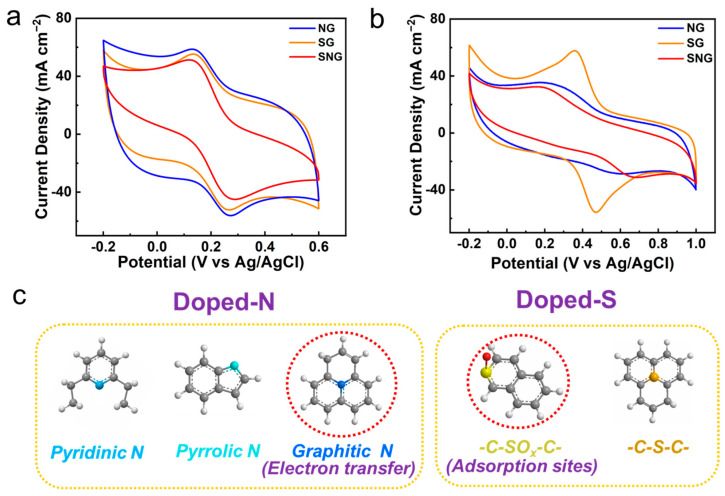
(**a**) The CVs of graphene samples in Fe (CN)_6_
^3−/4−^ + KCl solution. (**b**) CVs of graphene samples in Fe^2+/3+^ (H_2_O) + H_2_SO_4_ solution. [Fe (CN)_6_
^3−/4−^] =1 mM in [KCl] = 0.1 M; [Fe^2+/3+^(H_2_O)] =1 mM in [H_2_SO_4_] = 0.5 M; scan rate: 50 mV s^−1^. (**c**) Schematic illustrates the functional groups for the doped-N and doped-S.

**Table 1 ijms-23-14967-t001:** Atomic content, N/C, and O/C atomic ratio by EA analysis of OG, NG, SG, and SNG.

Samples	OG	NG	SG	SNG
C (at%)	13.6	48.6	50.1	49.6
H (at%)	23.1	33.4	23.6	15.9
O (at%)	63.3	13.7	25.1	22.2
N (at%)	-	4.3	-	7.9
S (at%)	-	-	1.2	7.5
N/C (at%)	-	8.9	-	6.3
S/C (at%)	-	-	2.4	6.6

**Table 2 ijms-23-14967-t002:** Different nitrogen and sulfur species by XPS analysis of NG, SG, and SNG.

Samples	NG	SG	SNG
pyridine nitrogen (%)	23.1	-	15.8
pyrrole nitrogen (%)	47.6	-	68.5
graphitic nitrogen (%)	29.3	-	15.7
-C-S-C- (%)	-	89.5	39.3
-C-SOx-C (%)	-	10.5	60.7

## Data Availability

Not applicable.

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
