# Peer review of "Identifying the Active Sites of Heteroatom Graphene as a Conductive Membrane for the Electrochemical Filtration of Organic Contaminants"

_ijms, 2022, doi:10.3390/ijms232314967_

Round 1
Reviewer 1 Report
The prepared manuscript shows and describes the influence of the graphene doping on the electrochemical filtration of two chosen organic contaminants. The paper seems to be interesting and it may add a few new informations to the literature. However, before publication the some issues need to be discussed:
1. FT-IR measurements - based on most papers in literature, in this region (~1020 cm-1) the signal comes from C-O bond (https://doi.org/10.1016/j.carbon.2019.12.011; https://doi.org/ 10.1039/C2JM33194B). C-N bonds are usually assigned at ~1250 cm-1 (same papers). S=O can be observed at ~1400 cm-1 (https://doi.org/10.1016/j.electacta.2019.05.015; https://doi.org/10.1002/cssc.201700910) and C-S is observed around ~800 cm-1. Please, revise obtained results and add relevant references to support your observations.
2. UV-vis studies - The shift of peak maximum by 3 nm is very small and I am not convinced that it can be assigned to successful doping of heteroatoms to graphene layers. Please prove your conclusion by using relevant references.
3. Page 4, line 100 - value 26.4% is different than value presented in Table 1.
4. Page 4, line 93 - "different materials were investigated by elemental analysis (EA) and XPS (Table 1)". Text is not consistent with the Table signature. Descripton of Table 1 suggests that in the table 1 are only results for EA analysis. Authors should point out which results are from EA and which are from XPS analysis.
5. Table 1 - Deconvoluted N 1s spectra shows three components, however the sum of those components in table 1 is not equal to 100% for NG and SNG samples.
6. Figure 2d - On the XPS survey spectrum of OG sample very small peak corresponding to nitrogen presence can be observed (~400 eV). Authors should explain the origin of nitrogen presence in undoped sample and the absence of nitrogen resulting from EA analysis.
7. XPS measurements - the references which was used to assigned the nitrogen and sulfur bonds to binding energy needs to be added to manuscript.
8. Synthesis of GO - Did the sample was washed before dispersion preparation for doping experiments? If yes, the short description of washing GO procedure should be added to manuscript.
Author Response
Thanks for your constructive suggestions, and please see the attachment.

Reviewer 2 Report
The manuscript titled “Identifying the active sites of heteroatom graphene as a conductive membrane for the electrochemical filtration of organic contaminants” written by Pan et al. is an interesting manuscript which describe the degradation of dye pollutants on foreign atom doped graphene.
It is well structured and written and the results are well presented. However, I have a few comments which needs to be addressed before I can recommend the acceptance in International Journal of Molecular Sciences.
1. The novelty of the work should be stated clearly. Has any previous similar work been done? If yes, this should be stated and how the current work differs or improves on the previous ones.
2. The figure quality needs to be improved. Especially figures 4 and 5.
3. A few typo is observed which needs to be corrected:
a. Line 83, “ where that of OG is around 275 nm of OG”. Statement not clear.
b. Line 239, “Graphite oxide (GO)”. Do you mean graphene oxide?

Author Response

(The authors gave the same response as above.)
